# *Arabidopsis* Restricts Sugar Loss to a Colonizing *Trichoderma harzianum* Strain by Downregulating *SWEET11* and *-12* and Upregulation of *SUC1* and *SWEET2* in the Roots

**DOI:** 10.3390/microorganisms9061246

**Published:** 2021-06-08

**Authors:** Hamid Rouina, Yu-Heng Tseng, Karaba N. Nataraja, Ramanan Uma Shaanker, Ralf Oelmüller

**Affiliations:** 1Department of Plant Physiology, Matthias Schleiden Institute of Genetics, Bioinformatics and Molecular Botany, Friedrich-Schiller-University Jena, 07743 Jena, Germany; a.h.rooina@gmail.com (H.R.); yu.tseng@uni-jena.de (Y.-H.T.); 2Department of Crop Physiology, University of Agricultural Sciences, GKVK, Bangalore 560065, Karnataka, India; nataraja_karaba@yahoo.com; 3School of Ecology and Conservation, University of Agricultural Sciences, GKVK, Bangalore 560065, Karnataka, India; umashaanker@gmail.com

**Keywords:** *Trichoderma*, SWEET11, SWEET12, SWEET2, SUC1, sucrose/sugar transporter, endophyte, symbiosis, *Arabidopsis*, phosphate starvation, *Trichoderma harzianum*

## Abstract

Phosphate (Pi) availability has a strong influence on the symbiotic interaction between *Arabidopsis* and a recently described root-colonizing beneficial *Trichoderma harzianum* strain. When transferred to media with insoluble Ca_3_(PO_4_)_2_ as a sole Pi source, *Arabidopsis* seedlings died after 10 days. *Trichoderma* grew on the medium containing Ca_3_(PO_4_)_2_ and the fungus did colonize in roots, stems, and shoots of the host. The efficiency of the photosynthetic electron transport of the colonized seedlings grown on Ca_3_(PO_4_)_2_ medium was reduced and the seedlings died earlier, indicating that the fungus exerts an additional stress to the plant. Interestingly, the fungus initially alleviated the Pi starvation response and did not activate defense responses against the hyphal propagation. However, in colonized roots, the sucrose transporter genes *SWEET11* and -*12* were strongly down-regulated, restricting the unloading of sucrose from the phloem parenchyma cells to the apoplast. Simultaneously, up-regulation of *SUC1* promoted sucrose uptake from the apoplast into the parenchyma cells and of *SWEET2* sequestration of sucrose in the vacuole of the root cells. We propose that the fungus tries to escape from the Ca_3_(PO_4_)_2_ medium and colonizes the entire host. To prevent excessive sugar consumption by the propagating hyphae, the host restricts sugar availability in its apoplastic root space by downregulating sugar transporter genes for phloem unloading, and by upregulating transporter genes which maintain the sugar in the root cells.

## 1. Introduction

Beneficial interactions between fungi and host plant roots rely on the exchange of nutrients between the symbiotic partners. Well investigated examples are mycorrhizal symbioses, where the plant delivers photo-assimilates to the fungi and fungi inorganic ions, in particular phosphate (Pi), and water to their hosts [1]. Quite similar exchanges might occur in beneficial symbiotic endophyte/plant interactions, although the molecular, cellular, and physiological bases are less understood [2,3] and ref. therein. All these beneficial symbioses are fragile and can shift to neutral, saprophytic, or even pathogenic interactions when stress conditions impair the exchange balance, and the survival of one of the symbiotic partners is compromised [4].

Upon Pi limitation, roots respond with the upregulation of Pi transporter genes and some Pi starvation regulators. The high affinity Pi transporters Pht1;1 and Pht1;4 play a major role in Pi acquisition under low Pi conditions [5]. *Pht1;1* and *-1;4* are induced by Pi starvation, and the transporter activities are controlled by additional mechanisms, including protein trafficking, localization, and degradation [6]. Furthermore, PHOSPHATE1 (PHO1) is the main Pi transporter into the xylem and *PHO1* is preferentially expressed in the root vascular system under low Pi [7]. Chen et al. [8] demonstrated that WRKY6 regulates *PHO1* expression, whereas low Pi reduced WRKY6-binding to the *PHO1* promoter, thereby stimulating the expression of the gene. In addition, expression of many Pi transporter genes is controlled by Pi starvation response (PHR) transcription factors [9,10,11,12,13]. Castrillo et al. [14] showed that PHR1 is a central regulator in balancing the Pi starvation response and plant immune regulation, which occurs mainly post-transcriptionally and is only weakly regulated at the transcriptional level.

Restriction of fungal growth due to nutrient limitations in the environment promotes colonization of the host plants by suppressing the host’s immune system. On the other hand, the increased demand of the fungus for photosynthates forces the plant to control its sucrose distribution to prevent loss to the microbes. Under Pi limitation, the plants transport more sugar from the leaves to the roots through the phloem to support root growth [15]. However, this also has tremendous effects on the growth of the root- associated fungi. They obtain the sugar either through direct transport from host cell to microbial cell [16], from the host apoplast, which contains the secreted sugar after release from the phloem cells, or from the rhizosphere, which contains secreted sugar from different root cells. Several transporter families have been proposed to transport sugars to the symbiotic partners [16]. Loading and unloading of the root phloem parenchyma cells with sugar involves the SWEET transporters [17,18] and ref. therein. The unloaded sucrose in the root apoplast is mainly transported into the mesophyll or epidermal root cells by SUC1 [2] and further transported into the vacuole of these cells by SWEET2, where it is stored [19]. Most of the hyphae of beneficial root-colonizing endophytic fungi grow in the root apoplastic space, which makes it likely that they also use the sugar from this plant compartment.

We studied the symbiotic interaction of *Arabidopsis* with a recently characterized *Trichoderma harzianum* strain [20]. The fungus strongly promotes plant growth on soil and full culture medium [20]. However, under Pi limitation, the fungus colonizes *Arabidopsis* aggressively, shifting the beneficial relationship towards an interaction with no or fewer benefits for the host. By analyzing genes for sugar and Pi transporters, as well as for defense, we were able to dissect the influence of Pi limitation on the symbiosis and propose a model for the shift in this interaction.

## 2. Methods and Materials

### 2.1. Growth Conditions of Plants and Fungus

*Arabidopsis thaliana* (ecotype Columbia-0) was used for this study. The seeds were surface-sterilized with a solution containing sterile distilled water (dH_2_O), sodium lauroyl sarcosinate, and DanKlorix (GP GABA GmBH Hamburg, Germany; 64%, 4%, 32%; *v/v/v*) for eight minutes under constant shaking, followed by six rinses with dH_2_O. Surface-sterile seeds were sown on MS media supplemented with 0.3% gelrite [21]. After cold treatment at 4 °C for 48 h, plates were incubated for 10 days at 22 °C under continuous illumination (100 µmol m^−2^ sec^−1^). The *Trichoderma* strain was propagated on KM medium (pH 6.5) for a week at 25 °C in the dark, as described previously, and pH changes were monitored by adding 0.004% (*w/v*) bromocresol purple pH indicator [20]. All experiments with insoluble Pi were performed on National Botanical Research Institute’s Pi growth medium (NBRIP) (glucose, 10 g/L: Ca_3_(PO_4_)_2_, 2.5 g/L; MgCl_2_ × 6 H_2_O, 5 g/L; MgSO_4_ × 7 H_2_O, 0.25 g/L; KCl, 0.2 g/L; and (NH_4_)_2_SO_4_, 0.1 g/L), either on plates with 0.3% gelrite or in liquid medium for fungal growth. The medium with soluble Pi, which was used as a control, contained the same molar Pi concentration of K_3_PO_4_ instead of Ca_3_(PO_4_)_2_.

### 2.2. Trichoderma-Arabidopsis Co-Cultivation

Cocultivation of *A. thaliana* and *Trichoderma* strain was performed under in vitro culture conditions. Ten-day-old *A. thaliana* seedlings of equal sizes were directly transferred from MS medium to NBRIP medium. The fungus was grown on NBRIP plates. The cocultivation of *Arabidopsis* plants and *Trichoderma* strain were performed on NBRIP medium. A 5 mm plug of the *Trichoderma* strain was placed at the center of the cocultivation’s plate. Cocultivation was monitored over a period of 10 days at 22 °C under continuous illumination (100 µmol m^−2^ s^−1^) [20,22].

### 2.3. RNA Extraction and cDNA Synthesis

RNA was isolated from the roots of colonized and uncolonized *Arabidopsis* seedlings with an RNA isolation kit (RNeasy, Qiagen, Hilden, Germany) and used for quantitative RT-PCR. Reverse transcription of 1 µg of total RNA was performed with an oligo(dT) primer. First strand synthesis was performed with a kit from Qiagen (Omniscript RT Kit, Qiagen, Hilden, Germany).

### 2.4. Real-Time PCR

Real-time quantitative RT-PCR was performed using the iCycler iQ real-time PCR detection system and iCycler software version 2.2 (Bio-Rad, Munich, Germany). For the amplification of the PCR products, iQ SYBR Supermix from Bio-Rad was used according to the manufacturer’s instructions in a final volume of 23 µL. The iCycler was programmed to 95 °C 3 min, 40 × (94 °C 30 s, 57 °C 30 s, 72 °C 40 s), 72 °C 10 min, followed by a melting curve program (50–85 °C in increasing steps of 0.5 °C). All reactions were repeated at least 3 times. The mRNA levels for each cDNA probe were normalized with respect to the *Arabidopsis glyceraldehyde-3-phosphate dehydrogenase 2* (*GAPC2*) and/or *ACTIN2* mRNA level. Fold induction values for the *tef1* mRNA levels (Genbank accession number: MT591352) from the *Trichoderma* cDNA were calculated with the ΔCt equation. Root colonization was determined relative to the plant *GAPC2* cDNA levels. The primer pairs for the *Arabidopsis* genes are given in the Appendix A.

### 2.5. Fluorescence Microscopy

The entire *Arabidopsis* roots, root tips, hypocotyls, and leaves were imaged using an AXIO Imager.M2 (Zeiss Microscopy GmbH, Jena, Germany) equipped with a 10× objective (N-Achroplan 10×/0.3). The bright field and fluorescence images (EX 545/25 and EM 605/70) were recorded with a color camera (AXIOCAM 503 color Zeiss, Jena, Germany) by use of an EGFP (EM 525/50 nm) and DsRED filter (EM 605/70 nm). Digital images were processed with the ZEN software (Zeiss, Jena, Germany), treated with Adobe R PhotoShop to optimize brightness, contrast, and coloring, and to overlay the photomicrographs to confirm fluorescence information.

### 2.6. Measurement of Photosynthesis Parameters

The *Arabidopsis* seedlings were dark-adapted for 15 min and the chlorophyll fluorescence was measured using a FluorCam 700F (Photon System Instruments, Czech Republic). Program parameters of FluorCam were set according to Wagner et al. [23]. The maximum quantum yield of the photosystem II (Fv/Fm) was calculated according to Maxwell and Johnson [24].

### 2.7. Acid Phosphatase (AcP) Assay

Extraction for the AcPase activity assays was carried out using 50 mL of sterilized NBRIP broth in a 100 mL conical flask. The flasks were inoculated with 1 mL of *Trichoderma* spore solution. The inoculated flasks were incubated at 25 °C for three days. The samples were centrifuged at 10,000 rpm for 10 min at 4 °C. The cell-free supernatant was assayed for crude AcP activity according to the method outlined by Tabatabai and Bremner [25]. One mL of the supernatant was mixed with 4 mL of modified universal buffer (pH 6.5). Further to this, 0.025 mM disodium p-nitrophenyl Pi (tetrahydrate) was mixed with the culture supernatant and incubated at 37 °C for 1 h. After 1 h of incubation, the reaction was stopped by adding 4 mL of 0.5 M NaOH and 1 mL of 0.5 M CaCl_2_. The concentration of p-nitrophenol was measured in triplicate by measuring the absorbance at 420 nm using a UV–Vis spectrophotometer, and the amount was quantified based on a standard curve with serially diluted solutions of p-nitrophenol. One unit (U) of phosphatase activity was defined as the amount of enzyme required to release 1 mol of p-nitrophenol/mL/min from di-Na p-nitrophenyl Pi (tetrahydrate) under the assay condition.

### 2.8. Quantitative Estimation of Soluble Pi

Erlenmeyer flasks containing 100 mL of NBRIP broth without bromophenol were inoculated with 1 mL spore solution of 5-day-old fungal cultures. Non-inoculated medium served as a control. The flasks were incubated in the dark at 25 °C for three days. The pH of the culture medium was measured 72 h after inoculation. A volume of 5 mL of the fungal culture was collected and centrifuged at 10,000 rpm for 10 min. The supernatant used to estimate the released Pi was determined spectrophotometrically (880 nm) according to Murphy and Riley [26].

## 3. Results

### 3.1. Pi Limitation Promotes Plant Colonisation

We have previously shown that cocultivation of *Arabidopsis* seedlings with the *Trichoderma* strain resulted in the colonization of the roots and the establishment of a beneficial symbiotic interaction. The fungus promoted growth of the host and defended it against pathogen attack [20]. Without stress, the fungal colonization was restricted to the roots, and hyphae were never detected in the aerial parts of the Arabidopsis seedlings [20]. To test how nutrient limitation influences the symbiotic interaction, we co-cultivated *Trichoderma* and *Arabidopsis* seedlings on NBRIP medium with insoluble Ca_3_(PO_4_)_2_ as sole Pi source.

As expected, when 10-day-old *Arabidopsis* seedlings were transferred from MS medium to the NBRIP medium, they started to die after an additional 10 days (Figure 1A). If these seedlings were transferred to soil, none of them recovered (n = 3 with 100 seedlings each; data not shown). Compared to control media with soluble Pi, the *Trichoderma* strain could only slowly grow in liquid NBRIP medium and on solid NBRIP agar plates (Figure 1B), since the fungus had AcP activity (Figure 1C) and can solubilize Ca_3_(PO_4_)_2_. The AcP activity, as well as solubilized Pi in the medium, is shown in Figure 1.

When *Arabidopsis* seedlings and the fungus were cocultured on NBRIY plates, we observed a fast root colonization (Figure 1D). After 2 days, hyphae can also be detected on the stem and, ultimately, leaves (Figure 2). Figure 2B shows the presence of hyphae at the stem/leaf junction 5 days after exposure to the fungus on insoluble Pi medium. At the same time, on soluble Pi medium, hyphae can only be detected in the roots (Figure 2B). The seedlings did not benefit from the interaction (Figure 1E). This can be shown by a recovery assay after transfer to soil. If the colonized seedlings were transferred to soil 8 days after colonization, 4.2 ± 0.6% (n = 3) survived, while 29.0 ± 3.7% (n = 3) survived when they were not exposed to the fungus.

The (*Thtef1*RNA)*/*(*AtGAPC2* RNA) ratio can be utilized to quantify fungal colonization. A full 24 h after cocultivation, *Thtef1* transcripts can be detected in the roots, but not in the shoots. Between 24 h and 72 h, we observed a greater than sixfold increase in root colonization (Figure 3). Between 72 and 96 h, leaf colonization increased strongly. This indicates that 72 h after the cocultivation, the root tissue is fully colonized, and additional colonization of the host requires propagation of the hyphae towards the aerial parts of the seedlings. Propagation of the hyphae to the aerial parts of the seedlings does not occur when the cocultivation was performed on media with soluble Pi (Figure 2B, cf. also [20]).

The effect of the fungus on plant performance under Pi stress was assessed by measurements of the efficiency of the photosynthetic electron transport (Fv/Fm) of photosystem II reaction center. A total of 48 h after transfer to Ca_3_(PO_4_)_2_-medium, the Fv/Fm value of the uncolonized seedlings was lower than that of the colonized seedlings. Following 96 h after transfer to Ca_3_(PO_4_)_2_-medium, this was reversed (Figure 4). This indicates that the plant initially benefits from the presence of the fungus, whereas, at the end of the experiments, this is no longer the case. The fungus tries to escape from the medium with insoluble Pi and prefers to grow on the host plant, despite the fact that it can solubilize Ca_3_(PO_4_)_2_.

### 3.2. Trichoderma Effects on the Pi Starvation Response in Roots

PHOSPHATE1 (PHO1) transfers Pi from root to shoot via Pi export into root xylem vessels. As expected, expression of this Pi starvation marker gene was upregulated in the roots after transfer of the seedlings to Ca_3_(PO_4_)_2_-containing medium (Figure 5). Interestingly, 48 h after the transfer, the *PHO1* mRNA level was higher in the roots of the colonized seedlings when compared to the uncolonized controls, whereas this was reversed 96 h after the transfer. A similar regulation was observed for the transcript levels of the high affinity Pi transporter Pht1;1 and Pht1;4 (Figure 5). This suggests that the fungus reduces the Pi starvation response during the early colonization phase, whereas the beneficial effect of the fungus is lost after 96 h. A possible explanation could be that the fungus initially provides Pi to the roots, either by direct transport from the fungal to the plant cell, or from the residual external soluble Pi. The mRNA level for PHR1 increases only marginally under Pi stress and root colonization (Figure 5), even though the transcription factor is considered as a central regulator controlling *PHT1* expression and Pi homeostasis in *Arabidopsis* ([27] and refs. therein). In addition, the *WRKY6* mRNA level does not change significantly in response to the applied Pi stress or fungal root colonization (Figure 5). In conclusion, and consistent with the photosynthetic parameters (Figure 4), it appears that, during the first period after transfer to the NBRIP medium, the colonized roots suffer less under Pi starvation than the uncolonized controls, as evident from the lower expression levels of *PHO1*, *Pht1;1*, and *Pht1;4*.

### 3.3. Fungal Colonization Does Not Activate Defense Gene Expression

The colonization of the roots, as well as aerial parts of the seedlings, by *Trichoderma* under Pi stress might be associated with defense gene activation. We checked classical marker genes for the jasmonate-dependent (*PDF1.2*) and salicylic-acid-dependent (*PR1*) defense pathways, but none of the investigated genes were upregulated more than twofold in the presence of the fungus (Figure 6). This suggests that the roots did not respond to the colonization by the activation of its defense machinery. Furthermore, abiotic stress often induces abscisic acid accumulation. However, the transcript level for the zeaxanthin epoxidase (ZEP) that functions in the first step of the biosynthesis of ABA was also not significantly regulated in response to Pi stress or root colonization (Figure 6). We conclude that the higher stress level of the *Arabidopsis* seedlings that were exposed to the fungus is not caused by more investment of the plant into defense against the aggressive fungal growth.

### 3.4. Root Colonization Alters Expression of SUC1, SWEET2, -11 and -12 in the Roots

Roots secrete a significant portion of sugars into the rhizosphere and deliver it to root-associated microbes. The increase in root colonization may influence the sugar distribution within the plant body, subsequently affecting plant survival. The sucrose transporter genes *SUC1* and *SUC2* are expressed in roots and their expression does not respond to Pi starvation in our experiment (Figure 7, Appendix A). However, at 48 and 96 h of Pi stress, we observed a greater than fivefold increase in the *SUC1* transcript level in the colonized roots, while the *SUC2* transcript level did not respond to the fungus (Figure 7, Appendix A). The monosaccharide transporter genes *STP1*, *STP13*, *ERD6*, and ERD6-like6 are also expressed in roots, but they did not respond to the Pi stress or root colonization (Appendix A). Among the SWEET sugar transporters, the *SWEET2* transcript level increased more than eight times in colonized roots (Figure 7), while there was no upregulation in the uncolonized controls. Furthermore, the *SWEET11* transcript level decreased more than tenfold, and that of *SWEET12* about sixfold 48 and 96 h after exposure to the fungus, but not in the uncolonized controls. *SWEET*3 can be used as a control, since it is expressed in roots but neither responds to the fungus, nor to the Pi stress (Figure 7).

Interestingly, the investigated *SUC*, *STP*, *ERD6*, and *ERD6-like6* transporter genes are also expressed in leaves, though to different extents than in roots, but we did not observe any significant response to the Pi stress or colonization (data not shown). Moreover, *SWEET11* and *-12* are expressed in leaves, although at much lower levels than in roots, and no regulation could be observed either (data not shown). The *SWEET2* transcript levels in the leaves are too low for meaningful interpretations.

These results show that *SUC1*, *SWEET2*, -*11*, and -*12* are specific targets of the fungus in roots under stress conditions.

## 4. Discussion

We established an artificial interaction system between *Arabidopsis* seedlings and a *Trichoderma* strain, which forces the fungus to escape from the growth medium and to colonize the roots, stems, and shoots of the seedlings. Without soluble Pi in the medium, the performance of the *Arabidopsis* seedlings is impaired and they ultimately die, which probably promotes the fast propagation of the fungus in the entire weakened host. The fungus can solubilize Ca_3_(PO_4_)_2_ and slowly grows on the medium, but prefers to escape to the plant.

One reason could be that it is easier for the fungus to withdraw Pi from the plant than solubilizing it from the medium. However, this appears to be unlikely, because we do not observe a stronger Pi starvation response of the plants when they are colonized by the fungus. In particular, during the first 2 days after cocultivation, the starvation response was even weaker, suggesting that the struggle for Pi is not the main reason for the colonization of the host.

Pi starvation triggers the colonization of the plant and propagation of the mycelium to the aerial parts, presumably because it is less energy-consuming for the fungus to grow on the plant than on the medium, and because the plant is weakened and does not activate processes to restrict fungal colonization. Without Pi stress, the colonization is restricted to the roots. It would be interesting to test whether other stresses will lead to similar strategies of the fungus.

However, the stronger colonization of the entire plant may force it to restrain the supply of reduced carbon to the fungus (cf. below). Therefore, our artificial system might mimic naturally occurring stress situations, in which beneficial plant/fungi symbiotic interactions in which only the roots are colonized by the microbes shift to more aggressive interactions which benefit the fungi. Since we did not observe a strong defense response of the weakened plant against fungal propagation, the interaction has more saprophytic than pathogenic features. Towards the end of the experiment, the plant dies and might provide material to the fungus which is used for their growth, similar or identical to saprophytic interactions. Several studies describe similar changes in fungal lifestyles. The establishment of a symbiotic interaction between *Arabidopsis* and *Colletotrichum tofieldiae*, which transfers Pi to the host and improves its growth under Pi limitation conditions, depends on the Pi availability, and Karandashov et al. [28] showed that the Pi stress causes a more saprophytic interaction. Grelet et al. [29] found that different *Mycena* species operate along a saprophytic–symbiotic continuum in association with *Ericaceae*. Intermediate interaction forms between endophytic and saprophytic lifestyles have also been proposed for wood-decaying fungi with different host species [30,31,32]; however, it is not clear how exogenous factors affect the interactions. An example for the adaptation to a symbiotic lifestyle provides *Archaeorhizomycetes* [33,34], which colonize the roots in summer and are mainly absent during the colder seasons. Schadt et al. [33] proposed that *Archaeorhizomycetes* might adjust their symbiotic lifestyle dependent on root-derived C compounds. Hacquard et al. [35] found genomic signatures in the transcriptomes of the beneficial root endophyte *Colletotrichum tofieldiae* and its pathogenic relative *C. incanum*, which are associated with the transition from pathogenic to beneficial lifestyles, and this included a narrowed repertoire of secreted effector proteins. Since we did not observe a strong defense response against the propagating fungal hyphae, nor a struggle for the limiting Pi source under our experimental conditions, access to sugar might be the main reason for the intense colonization. Like ectomycorrhia fungi, endophytes mainly colonize the apoplastic space within the root and live on the root surface. Therefore, most of the sugar required for the hyphal growth should be taken up from the plant apoplast.

### Trichoderma Manipulates Sugar Transport in Roots

The unloading of sucrose from the phloem in the roots occurs through a combination of symplastic and apoplastic pathways [36]. Cells in the root tip are connected to the phloem by plasmodesmata and form a symplastic domain [37,38]. In more mature areas of the root, cellular connections to the phloem are more limited, and transmembrane transport may be more important [39].

Apoplastic sugar unloading of the phloem in the roots is mainly mediated by *SWEET11* and *-12* in *Arabidopsis* roots ([18,40] and refs. therein). The higher demand for sugar due to the propagation of the hyphae results in the stimulation of photo-assimilate translocation from source to sink. In order to restrict sugar loss from the apoplast after unloading of the phloem, the plant downregulates *SWEET11* and -*12* (Figure 8). The involvement of SWEET transporters in beneficial and pathogenic plant/microbe interactions in roots is well-documented. Several *SWEET genes in Brasica crops* were significantly upregulated upon *P. brassicae* infection, the causal agent of clubroot [41]. Clubroot disease stimulates early steps of phloem differentiation and recruits SWEETs within developing galls [42]. Furthermore, MtSWEET11 is a nodule-specific sucrose transporter in *Medicago truncatula* [17,43,44].

SUC1 from *Arabidopsis* was one of the first H^+^-coupled sucrose transporters identified in plants [45,46], and, together with SUC2, is the most important sugar uptake carrier from the apoplast into the root cells [47,48]. Consistent with this, in *Arabidopsis* seedlings, *SUC1* expression is found in the elongation zone of the root and less in the root cap or zone of cell division, and *SUC1* expression in roots is induced by exogenous sucrose [48]. The most likely function for plasma-membrane-localized SUC1 [48] is in the uptake of sucrose released into the apoplast by SWEET sucrose efflux transporters as part of the phloem unloading mechanism [2]. *SUC1* expression is upregulated by *Trichoderma*, but not by Pi stress, suggesting that the roots compete with the fungus for the apoplastic sucrose.

Removal of the sugar from the apoplast by stimulating SUC1-mediated uptake into the root cells restricts fungal growth (Figure 8). Interestingly, the response to the fungus is mainly observed for *SUC1*, but not for *SUC2* (Figure 7 and Appendix A). Recently, Lasin et al. [49] demonstrated that *SUC1* introns act as strong enhancers of expression. The authors showed that a *SUC1* whole-gene *GUS* construct expressing a nonfunctional SUC1 mutant, that is transport inactive, is defective in sucrose-induced *SUC1* expression when expressed in an *suc1* knock-out mutant. The results indicate that sucrose uptake via SUC1 is required for sucrose-induced *SUC1* expression, and that the site for sucrose detection is intracellular [49]. Therefore, it is likely that the root cell recognizes sucrose shortage due to the propagating hyphae in its apoplast, and responds to it by stimulating the uptake and upregulation of *SUC1* (Figure 8).

Besides *SUC1*, *SWEET2* is strongly upregulated in the stress-exposed colonized roots. SWEET2 is located in the tonoplast of the mesophyll and epidermal root cells, and Chen et al. [19] proposed that the transporter modulates sugar secretion, possibly by reducing the availability of glucose sequestered in the vacuole, thereby limiting carbon loss to the rhizosphere. The reduced availability of sugar in the rhizosphere contributes to resistance to *Pythium*. Consistent with their interpretation, SWEET2 might be involved in reducing the available sucrose by transporting it into the vacuole of the mesophyll cells (Figure 8). Several studies show that root infections by pathogenic microbes alter the expression of plant sugar transporters to restrict sugar loss to the microbes; for instance, *SWEET* genes are upregulated in pathogenic interactions with bacteria and fungi [50,51,52,53]. SWEET11 is involved in disease resistance against the obligate biotrophic protist *P. brassicae*, the causal agent of clubroot [41]. Chen et al. [19] showed that the root-expressed vacuolar SWEET2 regulates sugar secretion, specifically from epidermal cells of the root apex, which influences growth of pathogenic microbes.

Vargas et al. [53] investigated *T. virens*-colonized roots and showed that sucrose exuded by plants also has a tremendous influence on the sucrose-dependent network in the fungal cells. The authors investigated fungal sucrose transporters and showed that they are also important for the symbiotic association. This study clearly highlights the necessity for further investigations on the role of sucrose in establishing and maintaining symbiotic plant/fungus interactions.

In summary, it appears 4 out of the 56 sugar transporters in *Arabidopsis* play an important role in the regulatory circuit. The proposed model (Figure 8) can now be tested with other endophytes and host/plant combinations, different stress conditions, and genetic tools manipulating the sugar transporter genes discussed in our model.

## Figures and Tables

**Figure 1 microorganisms-09-01246-f001:**
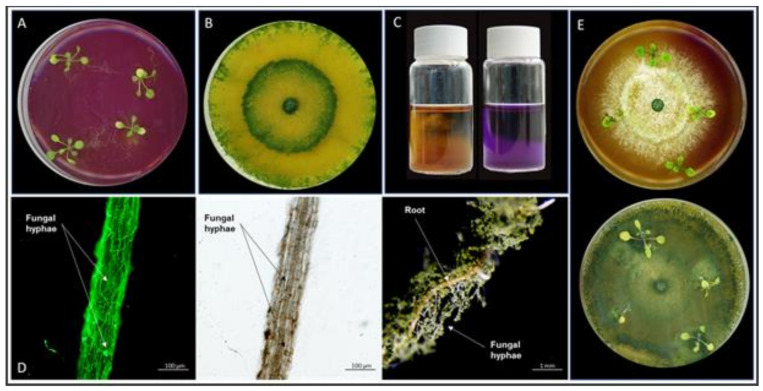
*Arabidopsis* and *Trichoderma* growth performance on NBRIP medium containing 0.004% (*w*/*v*) bromocresol purple as pH indicator (bright yellow indicates pH 5.2 and deep purple pH 7.0). (**A**) *Arabidopsis* seedlings started to die 10 days after cultivation on NBRIY medium. (**B**) Pi solubilization by *Trichoderma* 10 days after cultivation. (**C**) pH change induced by the fungus in insoluble Pi liquid medium. Left: directly after inoculation; right: 10 days later. (**D**) WGA Alexa 488 fluorescence staining of fungal hyphae after 10 days of coculture with *Arabidopsis* roots on NBRIP medium. Bright filed confocal (left) and dissection (right) images. (**E**) Cocultivation of *Arabidopsis* and *Trichoderma* for 5 days (top) and 10 days (bottom).

**Figure 2 microorganisms-09-01246-f002:**
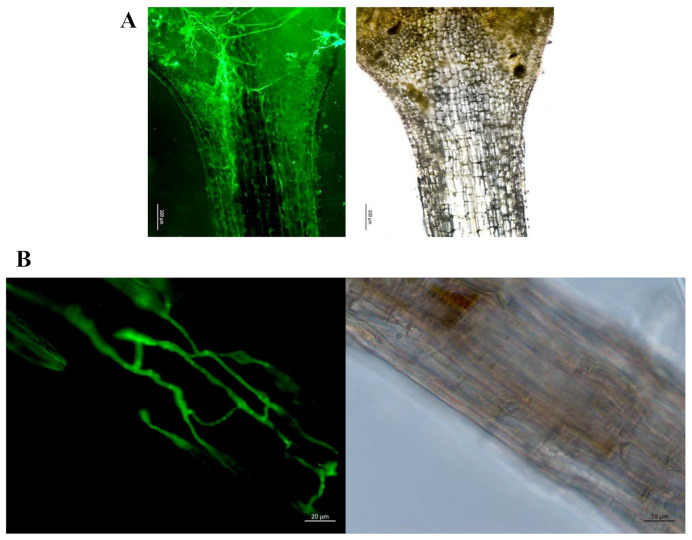
(**A**) WGA Alexa 488 fluorescence staining of *Trichoderma* hyphae at the stem/leaf junction of *Arabidopsis* seedlings, five days after cocultivation and transfer to NBRIP medium. (**B**) At the same time, after transfer to the medium with soluble phosphate, hyphae can only be detected in the roots, but never in the stems or leaves. Bright filed confocal (left) and dissection (right) images of roots. For experimental details, cf. Methods and Materials.

**Figure 3 microorganisms-09-01246-f003:**
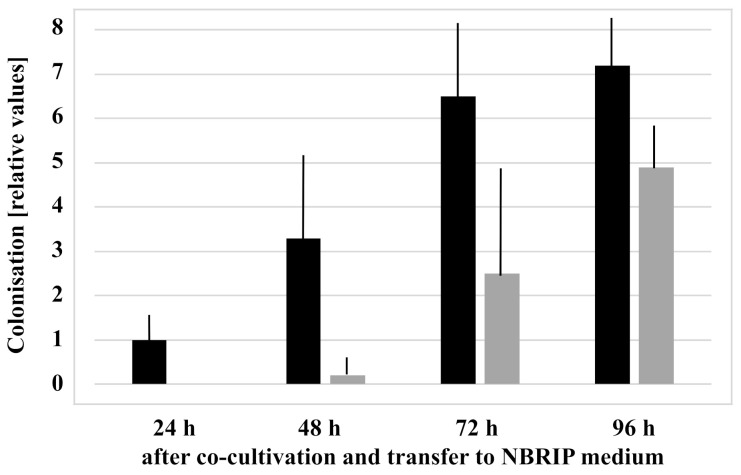
Colonization of *Arabidopsis* roots (**black**) and shoots (**grey**) by *Trichoderma* between 24 and 96 h after cocultivation on NBRIP medium. The amount of the fungus in the plant tissue was determined as the mRNA ratio (*Thtef1/AtGAPC2*) (cf. Methods and Materials). For better comparison, the value for the colonization of the roots 24 h after the onset of the experiment was set as 1.0, and all other values are expressed relative to this. Data are based on 3 independent experiments, bars represent SEs.

**Figure 4 microorganisms-09-01246-f004:**
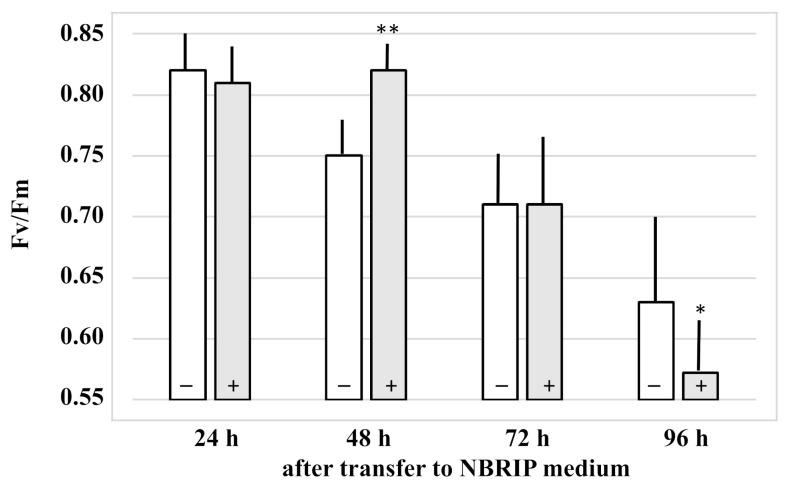
Maximum quantum yield of the photosystem II (Fv/Fm) after the transfer of *Arabidopsis* seedlings to NBRIP medium. At the time point of transfer to the NBRIP medium, the seedlings were either exposed to *Trichoderma* (+) or mock-treated (−). Data are based on 5 independent experiments, bars represent SEs. * indicates significant difference of *Trichoderma*-treated vs. non-treated roots (* *p* < 0.05; ** *p* < 0.005).

**Figure 5 microorganisms-09-01246-f005:**
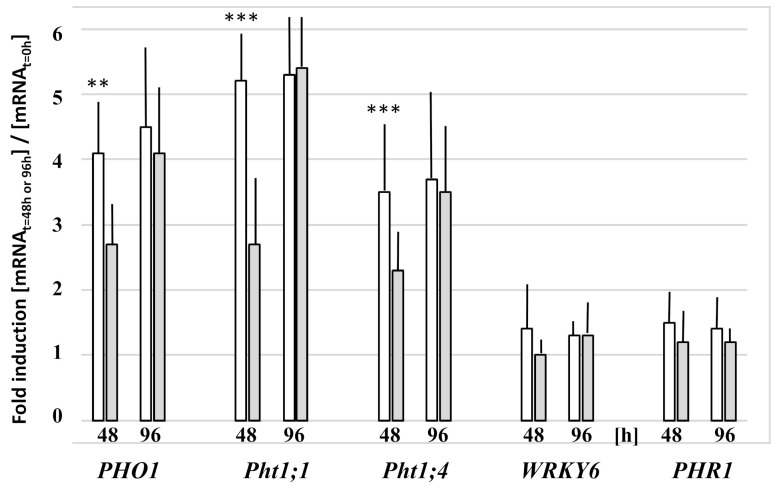
Relative mRNA levels for the Pi transporters Pht1;1 and Pht1;4, as well as the Pi starvation regulators PHO1, WRKY6, and PHR in the roots of *Arabidopsis* seedlings after transfer to NBRIP medium for 48 h or 96 h. Upon transfer to the NBRIP medium, the seedlings were either cocultured with *Trichoderma* (**grey**) or mock-treated (**white**). The mRNA levels are expressed relative to the levels in the roots at the time point of transfer to NBRIP medium (t = 0 h). Based on 3 independent experiments; bars represent SEs. * indicates significant difference of non-treated vs. *Trichoderma*-treated roots (** *p* < 0.005; *** *p* < 0.001).

**Figure 6 microorganisms-09-01246-f006:**
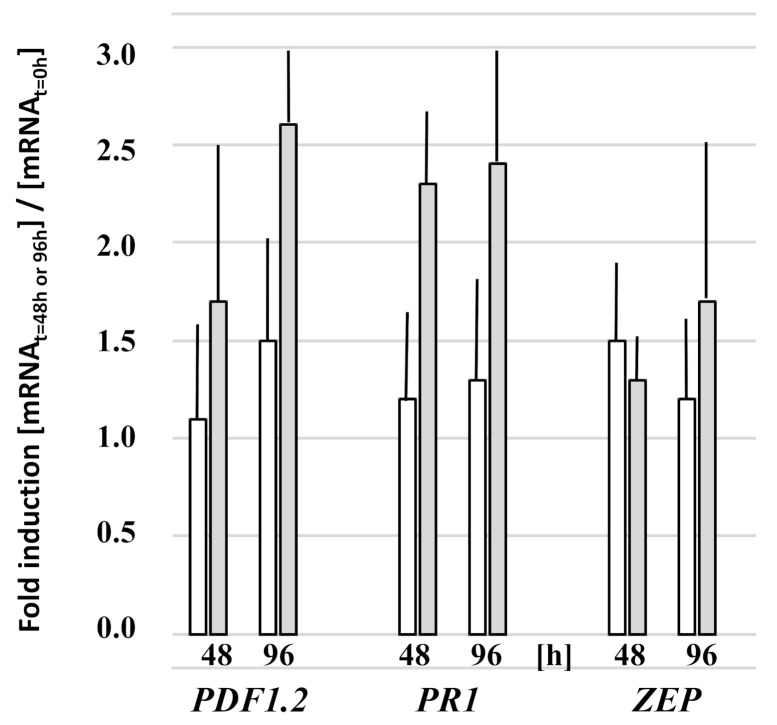
Relative mRNA levels for the defense-related proteins PDF1.2, PR1, and ZEP in the roots of *Arabidopsis* seedlings after transfer to NBRIP medium for 48 h or 96 h. Upon transfer to the NBRIP medium, the seedlings were either cocultured with *Trichoderma* (**grey**) or mock-treated (**white**). The mRNA levels are expressed relative to the levels in the roots at the time point of transfer to NBRIP medium (t = 0 h). Based on 3 independent experiments; bars represent SEs. No significant differences between the values of *Trichoderma*-treated vs. non-treated roots.

**Figure 7 microorganisms-09-01246-f007:**
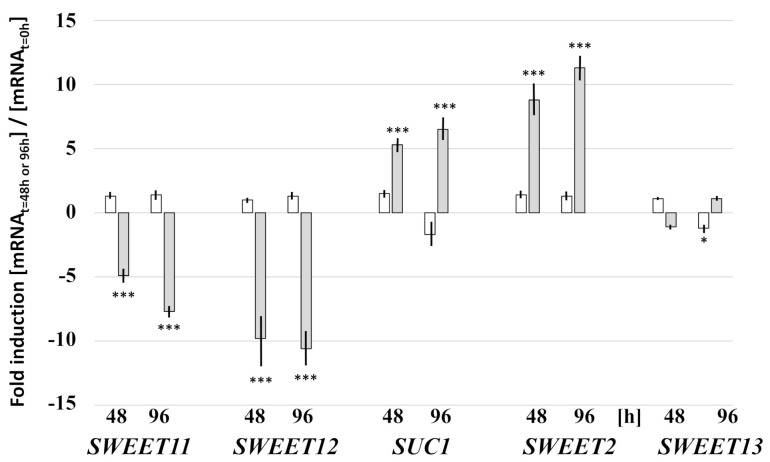
Relative mRNA levels for sugar transporter genes in the roots of *Arabidopsis* seedlings after transfer to NBRIP medium for 48 h or 96 h. Upon transfer to the NBRIP medium, the seedlings were either cocultured with *Trichoderma* (grey) or mock-treated (white). The mRNA levels are expressed relative to the levels in the roots at the time point of transfer to NBRIP medium (t = 0 h). Based on 3 independent experiments; bars represent SEs. * indicates significant difference of *Trichoderma*-treated vs. non-treated roots (* *p* < 0.05; *** *p* < 0.001).

**Figure 8 microorganisms-09-01246-f008:**
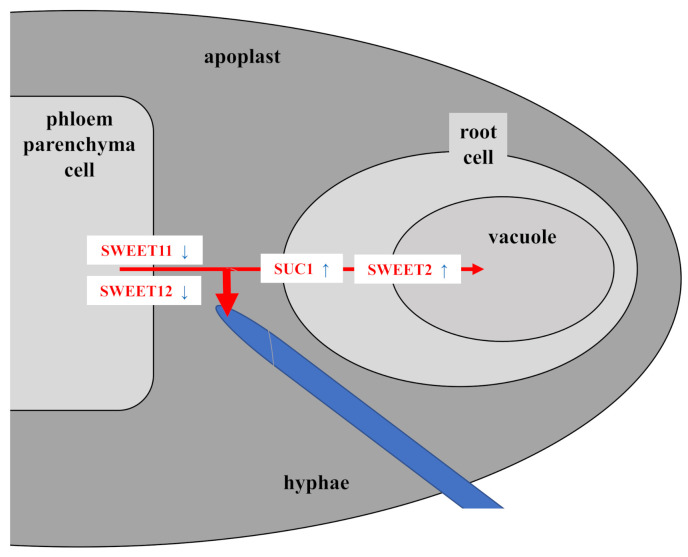
A model describing the regulation of sugar transporter genes in stress-exposed *Arabidopsis* roots in response to *Trichoderma* colonization. Upon increasing stress, the initial beneficial symbiotic interaction of *Trichoderma* with *Arabidopsis* roots shifts to an interaction with a saprophytic lifestyle of the fungus and a higher demand for sugar from the host. The endophytic fungus takes the sucrose mostly from the root apoplast, or from the rhizophere (not shown) after secretion of the roots. The host responds to it by downregulating *SWEET11* and -*12*, which results in less unloading of sucrose from the phloem into the apoplastic space, and thus restriction of hyphal growth due to sugar shortage. Restricted sucrose availability in the apoplast stimulates uptake into the mesophyll root cells and upregulation of *SUC1* expression to promote sucrose uptake from the apoplast into the mesophyll root cells. Furthermore, the shortage of sucrose in the root cells stimulates *SWEET2* expression. *SWEET2* sequesters sucrose in the vacuole of the root cells, to further reduce sugar loss to the microbe.

## Data Availability

Not applicable.

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
