# Peer review of "Arabidopsis* Restricts Sugar Loss to a Colonizing *Trichoderma harzianum* Strain by Downregulating *SWEET11* and *-12* and Upregulation of *SUC1* and *SWEET2* in the Roots"

_microorganisms, 2021, doi:10.3390/microorganisms9061246_

Round 1
Reviewer 1 Report
The manuscript entitled “Arabidopsis restricts loss to a colonizing Trichoderma harzianum strain by downregulating SWEET11 and -12 and upregulation of SUC1 and SWEET2 in the roots” have studied the symbiotic interaction of Arabidopsis with a recently characterized T.harzianum strain and analyzing genes for sugar and Pi transporters as well as for defense. Finally, Authors proposed a model describing the regulation of sugar transporter genes in stress-exposed Arabidopsis roots in response to Trichoderma colonization. Manuscript is interesting an I did not have any suggestion.
The study was written carefully and well in terms of language. In my opinion manuscript should be accepted.
Author Response
no comments required
Reviewer 2 Report
This Ms seems well done, but I have a few minor comments
line 81: Is it clorix or clorox?
line 86: pH value missing
line 195: field misspelled.
Suggest adding the following reference
1. Vargas W, Crutcher F, Kenerley C: Functional characterization of a plant-like sucrose transporter from the beneficial fungus Trichoderma virens. Regulation of the symbiotic association with plants by sucrose metabolism inside the fungal cells. New Phytol 2010, 189:777-789.
Author Response
the few comments are answered, cf. uploaded file.
RO
